# PCOS: A Chronic Disease That Fails to Produce Adequately Specialized Pro-Resolving Lipid Mediators (SPMs)

**DOI:** 10.3390/biomedicines10020456

**Published:** 2022-02-16

**Authors:** Pedro-Antonio Regidor, Xavier de la Rosa, Anna Müller, Manuela Mayr, Fernando Gonzalez Santos, Rafael Gracia Banzo, Jose Miguel Rizo

**Affiliations:** 1Exeltis Healthcare, Adalperostr. 84, 85737 Ismaning, Germany; anna.mueller@exeltis.com (A.M.); manuela.mayr@exeltis.com (M.M.); 2OTC Chemo, Manuel Pombo Angulo 28-4th Floor, 28050 Madrid, Spain; josemiguel.rizo@chemogroup.com; 3Center for Experimental Therapeutics and Reperfusion Injury, Department of Anesthesiology, Perioperative and Pain Medicine, Brigham and Women’s Hospital and Harvard Medical School, 60 Fenwood Road Boston, Boston, MA 02115, USA; xdelarosa@bwh.harvard.edu; 4Solutex SA, Avenida de la Transición Espanola 24, 28108 Alcobendas, Spain; fgsantos@solutexcorp.com; 5Solutex GC SL, Parque Empresarial Utebo, Avda. Miguel Servet n° 81, 50180 Utebo, Spain; rafael.gracia@insercolab.com

**Keywords:** PCOS, obesity, inflammation, specialized pro-resolving mediators (SPMs)

## Abstract

Introduction: Polycystic ovary syndrome (PCOS) is an endocrinological disorder that affects 5–15% of women of their reproductive age and is a frequent cause of infertility. Major symptoms include hyperandrogenism, ovulatory dysfunction, and often obesity and/or insulin resistance. PCOS also represents a state of chronic low-grade inflammation that is closely interlinked with the metabolic features. “Classical” pro-inflammatory lipid mediators such as prostaglandins (PG), leukotrienes (LT), or thromboxanes (TX) are derived from arachidonic acid (AA) and are crucial for the initial response. Resolution processes are driven by four families of so-called specialized pro-resolving mediators (SPMs): resolvins, maresins, lipoxins, and protectins. The study aimed to establish lipid mediator profiles of PCOS patients compared to healthy women to identify differences in their resolutive and pro-inflammatory lipid parameters. Material and Methods: Fifteen female patients (18–45 years) were diagnosed with PCOS according to Rotterdam criteria, and five healthy women, as a comparator group, were recruited for the study. The main outcome measures were: pro-inflammatory lipid mediators (PG, LT, TX) and their precursor AA, SPMs (resolvins, maresins, protectins, lipoxins), their precursors EPA, DHA, DPA, and their active biosynthesis pathway intermediates (18-HEPE, 17-HDHA, 14-HDHA). Results: The level of pro-inflammatory parameters in serum was significantly higher in PCOS-affected women. The ratio (sum of pro-inflammatory molecules)/(sum of SPMs plus hydroxylated intermediates) reflecting the inflammatory state was significantly lower in the group of healthy women. Conclusion: There is a strong pro-inflammatory state in PCOS patients. Further research will clarify whether supplementation with SPMs or their precursors may improve this state.

## 1. Introduction

Polycystic ovary syndrome (PCOS) is a disease that causes irregular bleeding, chronic anovulation, androgen excess, and a typical ovarian ultrasound feature [1]. It affects between 5 and 10% of women in their reproductive age, thus representing one of the most frequent causes of infertility [2]. The reasons for the development of PCOS have not been resolved yet. Genetic predisposition, together with the gestational environment and lifestyle factors, seem to be critical contributors [3]. Apart from the cardinal diagnostic criteria, including hyperandrogenism, ovulatory dysfunction, and/or the morphology of polycystic ovaries, as defined by the so-called “Rotterdam criteria” [4], other characteristics are related to the disease. PCOS is often accompanied by obesity [5], and 30–40% of women with PCOS show a reduced glucose tolerance, often accompanied by insulin resistance [6]. In total, 80% of obese women and 30–40% of lean individuals with PCOS suffer from hyperinsulinemia [6,7]. It has been found that hyperinsulinemia is a crucial factor in the clinical pathogenesis of PCOS and seems to be independent of weight [8]. Excess insulin may lead to enhanced androgen synthesis by direct stimulation of androgen production on the one hand and by reducing the serum levels of sex hormone-binding globulin (SHBG) on the other, thereby contributing to the androgen excess characteristic for PCOS [8]. In addition, obesity has a substantial impact on the severity of PCOS symptoms [9]. Apart from reinforcing insulin resistance, adipocytes show an altered hormone metabolism that contributes to the endocrinological disorder [10].

Furthermore, excess adipose tissue is a source of chronic low-grade inflammatory processes, and PCOS is considered an inflammatory disease [11]. Inflammatory response has been defined as an ensemble of initiation and active resolution processes. Within this perception, the resolution of inflammation is dependent on a class of lipid mediator molecules called specialized pro-resolving mediators (SPMs) [12]. These molecules are derived from polyunsaturated fatty acids eicosapentaenoic acid (EPA) and docosahexaenoic acid (DHA) and are synthesized via specific intermediate molecules by cells of the immune system. This publication will focus on the role of SPMs in chronic inflammatory diseases such as PCOS and the potential benefit of supplementation with their precursor molecules.

### Obesity, Insulin Resistance, and Inflammation

A state of chronic systemic inflammation is characteristic of obesity. It can be determined by measuring increased serum levels of inflammatory cytokines and altered frequencies and functions of peripheral blood lymphocytes [13,14,15]. These changes are manifested at the tissue level of the adipose, liver, and other tissue beds [14,15]. They might be responsible for comorbidities often related to obesity, such as atherosclerosis, diabetes, and steatohepatitis [16,17,18,19,20]. This kind of inflammation is often attributed to irregularities in innate immunity. However, innate and adaptive immune systems are closely interlinked, and consequently, obesity-related inflammation is associated with both processes [21]. For example, in obese individuals, systemic levels of free fatty acids are elevated [10]. These molecules are primary ligands of Toll-like receptors, which are critical regulators of the innate immune response [22,23]. In this way, the systems, which regulate obesity and inflammation, are linked directly.

A further relationship between inflammation and the metabolic system is visible on the cellular level since adipocytes and macrophages are closely related. Their evolution might be traced back to a conventional primordial precursor cell [24].

It has also been demonstrated that insulin resistance and inflammatory processes are closely linked and may stimulate each other [25]. Both subclinical inflammation and insulin resistance are important markers for the development of cardiovascular disease [26]. For women with PCOS, whose cardiovascular risks are elevated, a connection between inflammation and their hormonal–metabolic features was shown [27].

Since obesity, insulin resistance, and inflammation, key features of PCOS, are correlated, it is worthwhile looking at the possible pathways of inflammation that accompany PCOS, considering the modern perception of inflammatory processes.

The present study was therefore conceived to describe the physiological status of the innate immune response and its resolution potential by blood profiling of eicosanoid parameters and pro-resolving mediators in the plasma and sera of patients suffering from PCOS, as this information is lacking to date, and the results were compared with a healthy group to describe the intensity of the pro-inflammatory lipid mediator’s exacerbation in the peripheral blood in these patients.

## 2. Material and Methods

### 2.1. Human Plasma and Serum of PCOS Patients

Fifteen PCOS patients were evaluated in this study. The Rotterdam classification was used to define a patient as a PCOS woman. A control group of five healthy patients was used for comparison. The healthy patients were probands with no evident clinical acute disease or known pathological anamnesis in the medical history. The BMI of both groups was similar.

All patients were recruited at the Lubos Klinik in Munich, Germany. Samples were obtained at 8 a.m. under fasting conditions. Table 1 shows the demographic data of the patients.

The whole study was performed following a protocol designed and conducted following the ethical principles that have their origin in the Declaration of Helsinki and are consistent with GCP and existing regulatory requirements. Institutional review board approval was obtained from the study site.

### 2.2. Ethical Approval

Ethical approval was obtained for the investigational center. The overall approval for the study was given on 17 July 2020, by the Ethical Committee of the Bayerischen Landesärztekammer; No.: 20056. Clinical Trial Registration: DRKS-ID: DRKS00022337. Date of registration: 29 June 2020.

### 2.3. Blood Sample Analyses

Blood samples of plasma and serum were obtained for each patient. Each of these samples was considered a monoreplicate.

After standard preliminary treatment, samples were stored at −80 °C until they were processed in the laboratory. They were all analyzed separately.

### 2.4. Lipid Mediator Extraction and Profiling (LC-MS/MS)

Lipid mediators were extracted from human plasma and serum samples following the solid-phase extraction (SPE) method described below. Internal labeled standards d_8_-5-HETE, d_5_-RvD2, d_5_-LXA_4_, d_4_-LTB_4_, and d_4_-PGE_2_ (500 pg each,) in 4 mL of methanol (Methanol Optima LC/MS Grade, Fisher Chemical) were added to each sample (plasma or serum, 1 mL) previously thawed on ice. These labeled standards were used for the amount determined and the calculations of the recovery of the lipid mediators. Next, the samples were placed at −80 °C for 30 min to allow the precipitation of proteins. The probes were centrifuged in the following working step (2000× *g*, 10 min, 4 °C). The supernatants were obtained from each sample, and SPE was carried out according to optimized and reported methods [28,29]. Samples were rapidly acidified to pH = 3.5 with 9 mL of acidic water (HCl) before loading onto SPE columns (100 mg, 10 mL, Biotage) and pH neutralized with 4 mL of MilliQ water, followed by a wash step with 4 mL of n-hexane. After, compounds were eluted with 9 mL of methyl format. Extracts from the SPE were brought to dryness under a gentle stream of nitrogen and immediately resuspended in methanol/water (50:50 vol/vol) (MeOH/Water Optima LC/MS Grade, Fisher Chemical, both) before injection into an LC-MS/MS system.

### 2.5. Targeted LC-MS/MS Acquisition Parameters

The LC-MS/MS system consists of a Qtrap 5500 (Sciex) equipped with a Shimadzu LC-20AD HPLC pump. A Kinetex Core-Shell LC-18 column (100 mm × 4.6 mm × 2.6 μm, Phenomenex) was kept in a column oven maintained at 50 °C. A binary eluent system of LC-MS/MS-grade water (A) (Fisher Chemical) and LC-MS/MS-grade methanol (Fisher Chemical) (B), both with 0.01% (*v*/*v*) of acetic acid, was used as mobile phase. LMs were eluted in a gradient program with respect to the composition of B as follows: 0–2 min, 50%; 2–14.5 min, 80%; 14.6–25 min; 98%. The flow rate was 0.5 mL/min.

The QTRAP 5500 was operated in negative ionization mode, using scheduled multiple reaction monitoring (MRM) coupled with the information-dependent acquisition (IDA) and an enhanced product ion (EPI) scan. Each LM parameter (CE, target retention time (RT), and specific Q1 and Q3 mass) was optimized according to reported methods [29,30]. To monitor and quantify LMs of interest, quantities were taken as areas under the peak. We used MRM with MS/MS matching signature ion fragments for each molecule (at least six diagnostic ions; <0.1 picograms was considered below the limit of detection) using published criteria [30]. The laboratory analyses were performed at Solutex GC SL.

### 2.6. Statistical Analyses

Quantitative measurements were presented as mean, standard error, minimum, and maximum. When indicated, outlier exclusion was calculated using default parameters ROUT (Q = 1%) from GraphPad Prism version 9.0.2, GraphPad Software, San Diego, CA, USA. Comparisons were consequently made using an unpaired one-tailed *t*-test.

All tests were performed with a one-tailed t-test, and statistical significance was considered at *p* < 0.05. We did not make any adjustments for multiple testing; thus, the results are explorative and descriptive.

A ratio between pro-inflammatory and pro-resolutive parameters was established to describe the physiology of both investigated axes (pro-inflammatory and pro-resolutive).

The proposed ratios have the purpose of seeking the overall balance/unbalance of interconnected metabolic routes and the overall status of the resolution of the immune response.

### 2.7. Evaluated Parameters

Fatty acids (EPA, DHA, ARA, DPA); SPM monohydroxylated-containing precursors (17-HDHA, 18-HEPE, 14-HDHA); SPMs (Resolvins: RvE1, RvD1, RvD2, RvD3, RvD4, RvD5; maresins: MaR1, MaR2; protectins: PD1, PDX; lipoxins: LXA_4_, LXB_4_). Eicosanoids: prostaglandins (PGE_2_, PGD_2_, PGF_2α_); thromboxanes (TxB_2_); leukotrienes (LTB_4_).

## 3. Results

In this observational study, we observe that the quantity of each parameter was detectable in the sera but not in the same way in the participants’ plasma.

After quantitation, summation of total ARA-derived pro-inflammatory mediators resulted in a statistically significant increase (*p* < 0.05) when comparing sera from PCOS patients with healthy subjects. These pro-inflammatory mediators include LTB4, PGD2, PGE2, PGF2, and TXB2, and values altogether were 100 times higher compared to those of healthy subjects (see Figure 1). Measured prostanoids, including PGD2, PGE2, and PGF2, all together were increased by 600% in serum from patients with PCOS compared to healthy subjects (Figure 1). Thromboxane TXB2 was also statistically significantly (*p* < 0.05) higher in the serum from patients diagnosed with PCOS as compared to healthy subjects, which may reflect that these patients could suffer from coagulopathies (Figure 1).

We next studied whether these patients might have a disbalance in SPM formation. Specifically, the ratios of total pro-inflammatory lipid mediators, including LTB4, PGD2, PGE2, PGF2, and TXB2 vs. total SPMs formed, were statistically significantly different between the test groups (Figure 1). We observed in serum that the ratio of complete pro-inflammatory lipid mediators to the summation of SPMs, including 14-HDHA, 17-HDHA, and 18-HEPE, was statistically higher for patients with PCOS than that observed in the serum of healthy subjects (*p* < 0.05). This finding suggested that infections could impair resolution mechanism(s) due to exacerbated inflammation.

We quantified the free-fatty-acid precursors of resolving mediators. We observed that DHA, DPA, EPA, and ARA were not statistically significantly higher in PCOS patients’ plasma and serum (Figure 2) as compared to those of healthy subjects. Interestingly, we observed that PCOS patients presented statistically more elevated amounts of the ratio pro-inflammatory parameters/SPMs, including the monohydroxylates in the serum compared to the plasma (Figure 3).

As shown in Figure 1, the mean value of total prostanoids in the serum was 30,000 pg/mL in healthy subjects and 60,000 pg/mL in PCOS patients (see Figure 1). When comparing the differences for the thromboxane values between the healthy subjects and the PCOS patients, statistically significant differences could also be observed. PCOS patients expressed significantly higher values than the healthy controls (see Figure 1).

Figure 4 (plasma) and Figure 5 (serum) show in a graphical way the heat maps of the metalipidinomic results.

Human plasma or serum was extracted using SPE and subject to targeted LC-MS/MS (see method above). Targeted LM and pathway markers were profiled. Each mediator was identified using published criteria obtained on their structure, including identification criteria of at least six characteristic diagnostic ions present in their MS-MS spectra. Representative screen captures of MS-MS enhanced product ion (EPI) spectra captured from the chromatographic regions of (A) LTB4, (B) PGD2, (C) PGE2, (D) PGF2α, (E) TXB2, (F) RvD1, (G) PD1, (H) MaR1, and (I) MaR2. Screen captures were taken using SCIEX OS software. Insets: Chemical structures and prominent fragmentations.

Table 2 depicts all the values in tabular form.

## 4. Discussion

This study described the significant disbalance between pro-inflammatory eicosanoid-derived lipid mediators and pro-resolutive markers in the serum of PCOS patients. PCOS represents a chronic inflammatory condition since classical indicators for an inflammatory response are present, such as increased values of IL-6, C-reactive protein, fibrinogen, and erythrocyte sedimentation rate [11,31]. The presented data support this concept, as the lipidome of PCOS patients is shifted towards the pro-inflammatory axis with an increase in pro-inflammatory prostanoid derivatives and an elevated ratio of (pro-inflammatory LM)/(the sum of SPMs and their hydroxylated precursors).

Some widespread diseases such as diabetes, cardiovascular disease, and obesity are associated with chronic inflammation [15,17,32]. These pathologies are strongly interlinked with diet, and the positive impact of a polyunsaturated fatty acid (PUFA)-rich diet on cardiovascular health is broadly accepted [33]. Both EPA and DHA show an anti-inflammatory effect, and in this context, their role as precursors for SPM biosynthesis has been discussed [34,35]. The crucial role of SPMs in such chronic inflammatory states has become evident throughout the past years, and the underlying molecular mechanisms are increasingly elucidated [12,36].

For the DHA-derived SPM protectin PD1 and its hydroxylated precursors, for example, a positive influence on the metabolism of fatty tissue was demonstrated, suggesting a potential role in the management of obesity [37]. For DHA-derived SPM RvD1, a molecular mechanism for its possible cardioprotective effect has been demonstrated: it can activate lipoxin A4/formyl peptide receptor 2 (ALX/FPR2), which serves as a sensor for the resolution of inflammation in the context of coronary heart disease. In animal experiments, ALX7FPR-null mice developed obesity, diastolic dysfunction, and showed reduced SPM- levels associated with an impaired resolution of inflammation after cardiac injury [38].

SPM biosynthesis is, of course, based on the abundance of its PUFA precursors. However, in several experimental setups, the SPM biosynthesis was disturbed by altered activities of the involved enzymes [37]. Depending on the affected enzyme, supplementation with the hydroxylated intermediates of the SPM biosynthetic pathways may be efficacious in those cases, as demonstrated in a setting with leukocytes of obese individuals that showed a profound deficiency in the biosynthesis of RvD. Incubating the leukocytes with 17-HDHA, the precursor of RvD, restored the SPM production [39].

Therefore, supplementation with DHA- or EPA-derived SPMs and their corresponding hydroxylated precursor metabolites 18-HEPA, 17-HDHA, and 14-HDHA may represent a promising approach to address the pathologic features often associated with PCOS. Hyperandrogenemia, obesity, and insulin resistance all aggravate each other and are all associated with chronic inflammatory processes.

Another possible approach is the combination of PUFAs with Myo-inositol and/or D-chiro inositol. These two molecules have been classified as insulin sensitizers and seem to adequately counteract several insulin-resistant metabolic alterations with a safe nutraceutical profile. Paul et al. [40] concluded that supplementation with these two molecules complement each other in their metabolic actions and act in synergy with other insulin-sensitizing drugs and/or nutraceuticals.

Laganà et al. [41] also described that insulin resistance causes a rise in free-fatty-acid (FFA) plasma levels due to an increased synthesis from the liver and increased mobilization from adipose tissue. The excess of free fatty acids (especially the derivatives of omega-6) seems to lead to insulin resistance by inactivation of key enzymes such as pyruvate dehydrogenase (PDH) or by decreasing glucose transport activity in the cells. Therefore, a possible combination between the mediators of omega-3 and inositol may play a new treatment approach for women with PCOS.

In the present study, an association with the lipid mediator profile was demonstrated that was significantly shifted towards the pro-inflammatory axis compared to healthy women. Exciting was the greatly increased level of thromboxane TXB2 in PCOS patients compared to the test group, as its precursor, TXA2, plays an essential role in platelet activation and aggregation. PCOS-affected women are known to have a 2-fold increased risk for venous thromboembolic events compared to healthy women.

Our study supports the former investigation of Duleva et al. [42] and Rudnika et al. [11] that could demonstrate using classical inflammatory markers that PCOS is an inflammatory disease.

With this study, we could reinforce the hypothesis that was formulated in a previous review by Regidor et al. [35]. Hence, it is important to mention that the metabolites of omega-3 fatty acids show profound anti-inflammatory activity, as they diminish the synthesis and action of pro-inflammatory mediators such as LTs, PGs, and PAF, stimulate the anti-inflammatory M2 phenotype of macrophages and increase the number of anti-inflammatory molecules such as IL-10, limit recruitment of neutrophils, trigger the macrophage switch to the anti-inflammatory M2 phenotype, and increase their phagocytotic and efferocytotic action, thus contributing to clearance of the site of inflammation. They are increasingly proposed for treatment of chronic inflammatory states such as cardiovascular disease or obesity and diabetes. As PCOS women are highly affected by these diseases, we suppose that SPMs will play an essential role in the future management of this clinical condition.

The fact that white blood cells, cytokines, and interleukins are elevated could be reaffirmed with our data showing a metalipidinomic disbalance in women with PCOS.

Treatment of PCOS is mainly focused on weight loss, use of anti-androgenic hormone preparations, treatment with insulin-sensitizing agents or the whole repertoire of ovulation induction, and in vitro fertilization techniques when child wish comes into focus. At the same time, the underlying inflammatory processes are relatively neglected as a therapeutic target [43].

## 5. Conclusions

As the SPMs, which derive from 18-HEPE, 17-HDHA, and 14-HDHA, and are biosynthesized from their respective precursor omega-3-fatty acids EPA and DHA, have a possible influence on the resolution of inflammation associated with polycystic ovary syndrome, targeted supplementation with SPMs or their precursors may be a valuable novel therapeutic strategy worth further investigation in the management of women with a PCOS.

This study was able to show the lack of SPMs in women combined with overexpression of pro-inflammatory substances and also thromboxanes. This finding could also explain the 2-fold higher risk of thromboembolic events that is associated with PCOS syndrome.

## Figures and Tables

**Figure 1 biomedicines-10-00456-f001:**
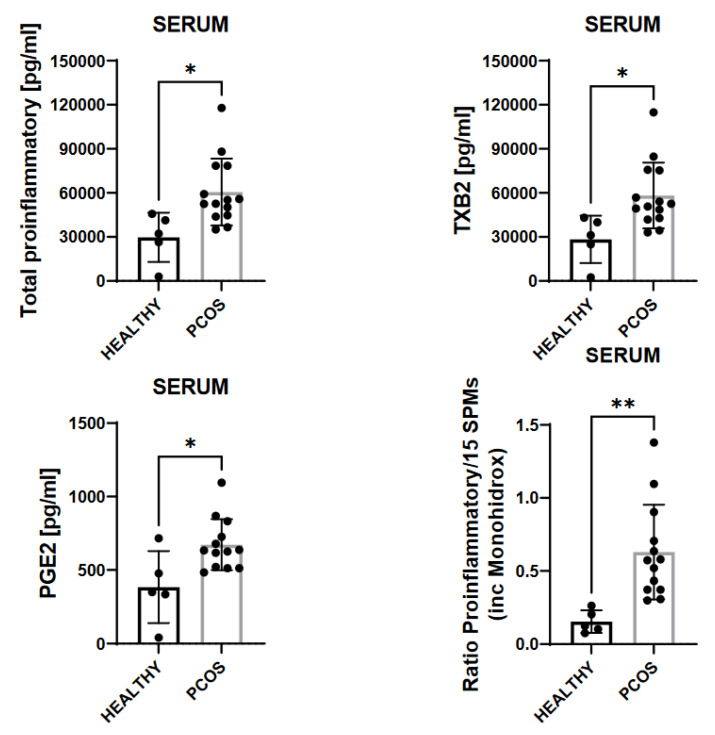
Results of the ARA-derived pro-inflammatory mediators of the healthy and PCOS patients. * = significant difference (*p* < 0.05); ** *p* < 0.005.

**Figure 2 biomedicines-10-00456-f002:**
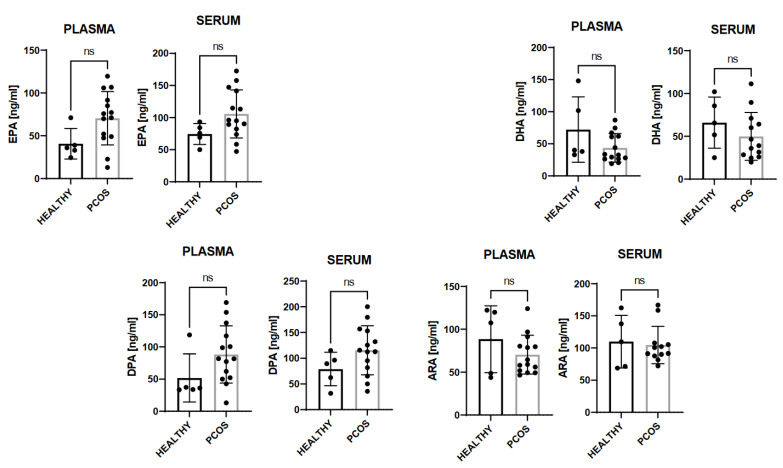
Results of the quantified free-fatty-acid precursors of resolving mediators. Ns = not significant.

**Figure 3 biomedicines-10-00456-f003:**
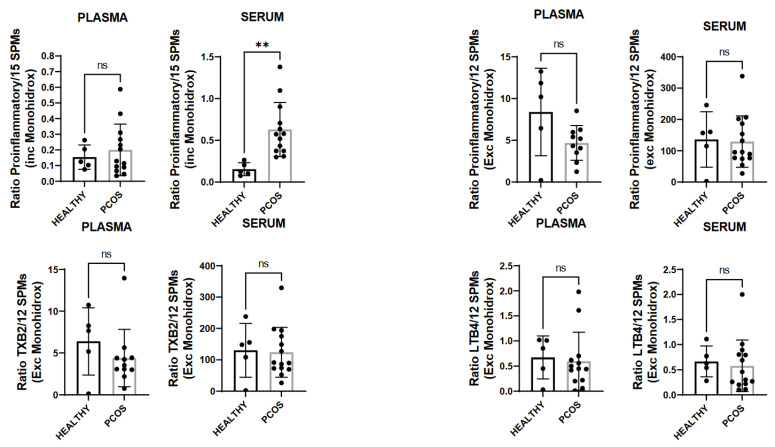
Results of the ratio pro-inflammatory parameters/SPMs, including the monohydroxylates in the serum compared to the plasma. ns = not significant; ** *p* < 0.005.

**Figure 4 biomedicines-10-00456-f004:**
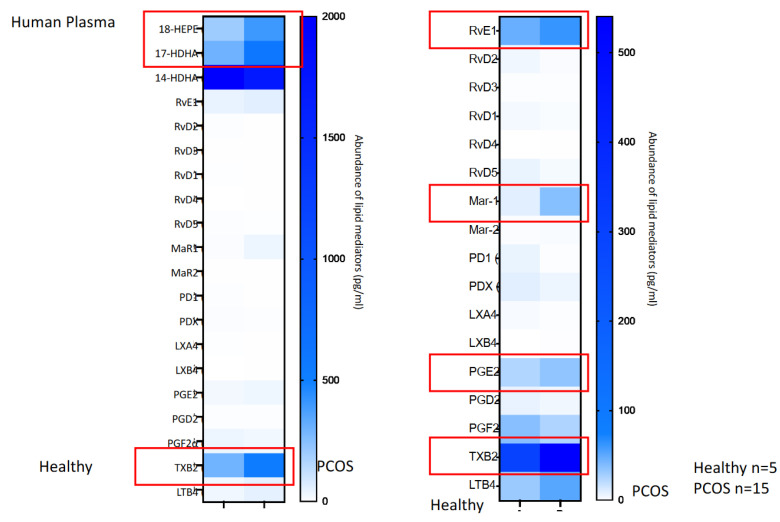
Heat map results in pg/mL of the human plasma analyses of the healthy and PCOs patients.

**Figure 5 biomedicines-10-00456-f005:**
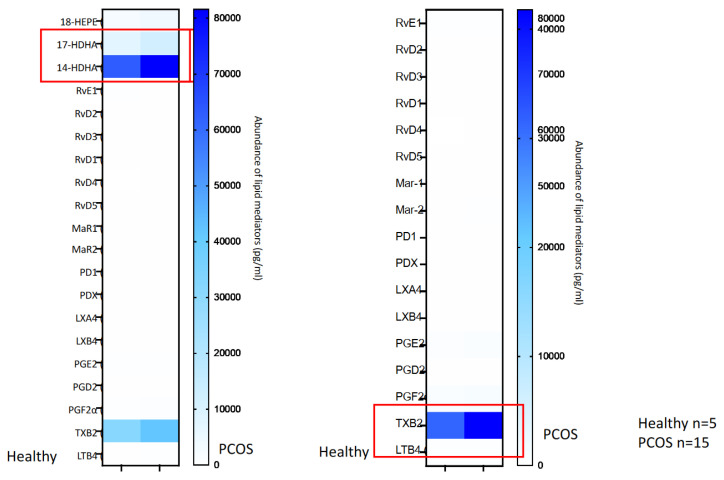
Heat map results in pg/mL of the human serum analyses of the healthy and PCOs patients.

**Table 1 biomedicines-10-00456-t001:** Demographical data of the study population.

	Healthy (*n* = 5)	PCOS (*n* = 15)
Sex	Female (5)	Female (15)
Age (years)	32 (29–34)	31 (24–42)
Weight (kg)	60.7 (50–76)	85.8 (54–110)
High (cm)	167.4 (162–173)	164.9 (150–175)
BMI (kg/m^2^)	21.5 (17.9–28.3)	31.7 (19.8–41.9)
PCOS	NA	+(15)

**Table 2 biomedicines-10-00456-t002:** Values in tabular form.

	PCOS	PCOS	Healthy	Healthy	PCOS	PCOS	Healthy	Healthy
	Plasma	Plasma	Plasma	Plasma	Serum	Serum	Serum	Serum
	Averages	SD	Averages	SD	Averages	SD	Averages	SD
Markers								
EPA (ng/mL)	70.1	29.1	42.7	15.2	103.6	35.8	77.0	14.5
DHA (ng/mL)	42.2	21.3	67.5	42.9	54.6	35.4	65.1	24.4
DPA (ng/mL)	89.0	41.5	57.4	33.0	113.8	44.7	85.0	29.7
ARA (ng/mL)	69.9	21.0	82.7	34.2	124.9	64.6	105.8	34.7
18-HEPE (ng/mL)	0.4	0.2	0.2	0.1	4.0	3.6	2.4	1.3
17-HDHA (ng/mL)	0.6	0.3	0.3	0.2	11.8	5.9	7.1	3.0
14-HDHA (ng/mL)	1.6	1.2	2.0	1.3	79.8	31.8	61.2	27.4
RvE1 (pg/mL)	97.6	254.0	39.1	87.4	188.4	317.1	170.4	381.1
RvD2 (pg/mL)	1.9	2.1	3.7	8.3	4.8	13.1	0.0	0.0
RvD3 (pg/mL)	1.1	0.9	0.7	1.7	0.7	1.4	0.9	1.4
RvD1 (pg/mL)	1.9	2.1	3.2	6.0	3.9	5.1	2.5	3.8
RvD4 (pg/mL)	0.3	0.6	0.0	0.0	1.1	2.4	0.0	0.0
RvD5 (pg/mL)	2.7	3.2	5.7	5.9	83.7	44.4	34.2	13.1
Mar-1 (pg/mL)	38.0	36.4	7.7	11.7	81.8	77.3	21.5	27.9
Mar-2 (pg/mL)	2.5	2.7	1.2	1.8	231.0	164.8	93.0	51.4
PD1 (pg/mL)	1.3	2.3	5.9	8.1	3.7	4.3	7.4	3.2
PDX (pg/mL)	7.5	9.2	7.4	10.4	72.1	33.1	41.6	28.7
LXA4 (pg/mL)	1.0	1.1	2.2	2.6	1.3	2.4	1.6	1.6
LXB4 (pg/mL)	2.0	5.8	0.0	0.0	2.0	5.6	2.7	6.0
PGE2 (pg/mL)	32.9	46.9	21.5	15.6	716.3	295.6	525.8	375.0
PGD2 (pg/mL)	5.2	4.1	6.1	4.1	58.9	18.0	51.8	20.1
PGF2α (pg/mL)	25.6	14.6	33.1	44.7	1275.9	578.1	777.4	474.8
TXB2 (pg/mL)	511.7	592.1	275.4	147.5	54,730	24,475	32,555	16,236
LTB4 (pg/mL)	51.3	90.4	28.0	14.8	273.6	160.4	211.5	74.7

## Data Availability

Not applicable.

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
