# Peer review of "PCOS: A Chronic Disease That Fails to Produce Adequately Specialized Pro-Resolving Lipid Mediators (SPMs)"

_biomedicines, 2022, doi:10.3390/biomedicines10020456_

Round 1

Reviewer 1 Report

The present paper describes in a very sound way the balance of pro-resolving mediators (SPMs) in the PCO patients. The finding also supports the already published data, however, makes it very clear that low grade inflammation in PCO is an important issue. 
Only minor suggestions for the manuscript:
I would be very helpful in the material and methods to declare which criteria was used. (I can indirectly see it is the Rotterdam criteria, but make it clear in methods).
Further please discuss a little more in details your findings in correlation to Regidor PA, Mueller A, Sailer M et al.

Author Response

I have introduced the changes that have been propoused.

Reviewer 2 Report

I read with great interest the manuscript, which falls within the aim of this Journal. In my honest opinion, the topic is interesting enough to attract the readers’ attention. Nevertheless, authors should clarify some points and improve the discussion, as suggested below.

Authors should consider the following recommendations:

  • Manuscript should be further revised in order to correct some typos and improve style.
  • Accumulating evidence suggests that one of the most important mechanisms of PCOS pathogenesis is the insulin-resistance. For this reason, the use of insulin-sensitizers, such as inositol isoforms, gained increasing attention due to their safety profile and effectiveness. Authors may better discuss this point, taking to account these recent articles: PMID: 26927948; PMID: 27579037

Author Response

The requested changes have been introduced.
